# Perioperative Systemic Treatment for Muscle-Invasive Bladder Cancer: Current Evidence and Future Perspectives

**DOI:** 10.3390/ijms22137201

**Published:** 2021-07-04

**Authors:** In-Ho Kim, Hyo-Jin Lee

**Affiliations:** 1Department of Internal Medicine, Division of Medical Oncology, Seoul St. Mary’s Hospital, The Catholic University of Korea College of Medicine, Seoul 06591, Korea; ihkmd@catholic.ac.kr; 2Department of Internal Medicine, Chungnam National University School of Medicine, Daejeon 35015, Korea

**Keywords:** bladder cancer, immunotherapy, perioperative systemic treatment

## Abstract

Radical cystectomy is the primary treatment for muscle-invasive bladder cancer; however, approximately 50% of patients develop metastatic disease within 2 years of diagnosis, which results in dismal prognosis. Therefore, systemic treatment is important to improve the prognosis of muscle-invasive bladder cancer. Currently, several guidelines recommend cisplatin-based neoadjuvant chemotherapy before radical cystectomy, and adjuvant chemotherapy is recommended in patients who have not received neoadjuvant chemotherapy. Immune checkpoint inhibitors have recently become the standard treatment option for metastatic urothelial carcinoma. Owing to their clinical benefits, several immune checkpoint inhibitors, with or without other agents (including other immunotherapy, cytotoxic chemotherapy, and emerging agents such as antibody drug conjugates), are being extensively investigated in perioperative settings. Several studies for perioperative immunotherapy have shown that immune checkpoint inhibitors have promising efficacy with relatively low toxicity, and have explored the predictive molecular biomarkers. Herein, we review the current evidence and discuss the future perspectives of perioperative systemic treatment for muscle-invasive bladder cancer.

## 1. Introduction

Bladder cancer is the 12th most common malignancy worldwide, causing approximately 200,000 deaths annually [1]. Muscle-invasive bladder cancer (MIBC) represents approximately 20% of newly diagnosed bladder cancer cases [2]. Currently, radical cystectomy (RC) with pelvic lymph node dissection is the primary treatment for MIBC; however, the disease tends to recur within two years in approximately 50% of patients [3]. Therefore, perioperative systemic treatment is important to improve MIBC prognosis. Current international guidelines recommend cisplatin-based neoadjuvant chemotherapy (NAC) followed by RC in patients with MIBC; adjuvant chemotherapy is also an option for select patients [4,5]. Recently, owing to the success of immunotherapy in treating metastatic disease, a perioperative immunotherapy-based treatment strategy for MIBC is being extensively investigated. Herein, we review the current evidence, and discuss the future perspectives of perioperative systemic treatment for MIBC.

## 2. Data Acquisition

A literature survey for current data on perioperative systemic therapy was conducted using the PubMed and ClinicalTrials.gov databases. The following combination of MeSH terms was used in the data searching process: “urothelial carcinoma”, “transitional cell carcinoma”, or “bladder cancer”; and “neoadjuvant”, “adjuvant”, “preoperative”, “postoperative”, or “perioperative”. Prospective/retrospective studies, systematic reviews, and meta-analyses were included. As perioperative immunotherapy-based treatment is a rapidly changing field, we also examined abstracts from major oncology conferences between January 2010 and March 2021. Moreover, we included the available data from ongoing clinical trials on perioperative immunotherapy.

## 3. Perioperative Chemotherapy in MIBC

Currently, cisplatin-based NAC followed by RC is the standard of care for MIBC. The National Comprehensive Cancer Network (NCCN) guidelines recommend cisplatin-based NAC as a category 1 treatment for patients with clinical T2–4a (cT2–4a) or N1 who are fit for cisplatin treatment [5]; the European Association of Urology (EAU) guidelines also strongly recommend cisplatin-based NAC for cT2–T4a disease [4]. Several randomized phase 3 studies have shown that NAC has a clinical benefit in MIBC [6,7,8]. In 2003, SWOG/The Eastern Cooperative Oncology Group and Cancer and Leukemia Group B reported the results of neoadjuvant methotrexate, vinblastine, doxorubicin, and cisplatin (MVAC) in 317 patients with cT2–4aN0M0 bladder cancer. The primary endpoint of the study was overall survival (OS). The median OS and 5-year survival rates were 77 months and 57%, respectively, in the NAC group vs. 46 months and 43%, respectively, in the surgery-alone group [6]. A Nordic collaborative group performed a combined analysis of two separate trials that were similar in design and had the same source population [7]; this study showed that NAC showed an 8% improvement in the 5-year OS rate (56% in the NAC group vs. 48% in the control group), and was associated with a 20% reduction in the relative probability of death. The BA06 30894 trial was an international, multicenter study that compared local radical treatment alone with neoadjuvant cisplatin, methotrexate, and vinblastine (CMV), followed by local radical treatment. The first analysis of this study showed a nonsignificant 15% reduction in the risk of death associated with neoadjuvant CMV [8]. In 2011, the updated analysis of this study, with a median follow-up of eight years, showed a statistically significant 16% reduction in the risk of death associated with neoadjuvant CMV, compared with the control group. Furthermore, the 10-year survival increased from 30% to 36% after neoadjuvant CMV, compared with the control group [9]. Three meta-analyses were performed to evaluate the clinical benefits of NAC; these studies confirmed that NAC led to an improvement in OS of approximately 5% in MIBC [10,11,12]. A modified MVAC has been investigated in two small, single-arm phase 2 trials [13,14]. Choueiri et al. (NCT00808639) evaluated the efficacy and safety of neoadjuvant dose-dense MVAC (dd-MVAC) with pegfilgrastim support in 39 patients with MIBC (cT2–cT4N0–1M0) [13]. The primary endpoint was pathologic response (PaR), defined by pathologic downstaging to ≤pT1N0M0. This study showed that 49% of patients achieved PaR, with 10% showing grade 3 or higher treatment-related toxicities. Plimack et al. (NCT01031420) assessed the feasibility of neoadjuvant-accelerated MVAC with pegfilgrastim in 44 patients with MIBC (cT2–cT4N0–1M0) [14]. The primary endpoint was pathologic complete response (pT0, pCR). This study revealed 38% pCR rates and 52% downstaging to non-muscle-invasive disease. Most patients (82%) experienced only grade 1–2 treatment-related toxicities. Combined gemcitabine and cisplatin (GC) treatment is also used in the neoadjuvant setting because of its similar OS and progression-free survival (PFS) and lower toxicity in metastatic disease compared with conventional MVAC [15]. Although GC has not been investigated in large-scale randomized prospective phase 3 trials, several retrospective and pooled analyses suggest that GC has a similar response in terms of pathologic downstaging to T0/1 [16,17,18,19,20]. The GETUG/AFU V05 VESPER trial (NCT01812369) is a randomized phase 3 controlled study comparing the efficacy of GC and dd-MVAC in a perioperative setting [21]. Among the 537 patients in the neoadjuvant group, pCR was observed in 36% of GC and 42% of dd-MVAC patients (*p* = 0.2); downstaging to organ-confined disease (<ypT3pN0) was achieved in 63% (GC) and 77% (dd-MVAC) of patients, respectively (*p* = 0.001). Grade 3 or higher hematologic toxicities were observed in 52% of patients in the dd-MVAC group and 55% of patients in the GC group. Grade 3 or higher gastrointestinal toxicity (*p* = 0.003) and asthenia (*p* < 0.001) were more frequently observed in the dd-MVAC arm. As this was a preliminary report, the result of the PFS as the primary endpoint has not yet been reported. We have summarized the results of NAC trials in MIBC in Table 1.

The role of adjuvant chemotherapy (AC) in MIBC has been investigated in several studies, although few studies are available as a reference. The EORTC 30994 trial (NCT00028756) was the largest AC trial that compared adjuvant versus deferred cisplatin-based combination chemotherapy after RC in patients with pT3–pT4 or N+ M0 urothelial carcinoma (UC) [22]. A total of 284 patients were randomly assigned (1:1) to either immediate AC (four cycles of GC, high dose MVAC, or conventional MVAC) or six cycles of deferred chemotherapy at relapse. The primary endpoint was OS. Although this study showed that AC significantly prolonged disease-free survival (DFS) compared with deferred treatment, no significant improvement in OS was noted with AC. The failure of this trial may be attributed to poor trial design and an inappropriate primary endpoint, and detailed information and the response of salvage treatment were not obtained. This explains why the observed benefit in DFS does not translate into a benefit in OS. Other trials [23,24,25,26] were in favor of AC; the clinical impression of these trials is limited, as these trials either did not achieve the primary endpoint or did not have a significant clinical implication because of poor study design, incomplete patient accrual, or early termination. In 2013, an updated meta-analysis was performed including 945 patients in 9 randomized clinical trials [27]. This study showed evidence of an OS and DFS benefit in patients with MIBC receiving AC after RC. Currently, the NCCN and EAU guidelines [4,5] recommend AC in patients with MIBC who have not received NAC; however, NAC is preferred over AC as the perioperative treatment option. We have summarized the trials for AC in patients with MIBC in Table 2.

## 4. Neoadjuvant Immunotherapy in MIBC

Recently, immune checkpoint inhibitor (CPI) therapy has become the standard treatment option for metastatic urothelial carcinoma (mUC) [28,29,30,31,32,33,34,35,36,37] (Table 3). In 2016–2017, atezolizumab, avelumab durvalumab, nivolumab, and pembrolizumab were approved for use in mUC by the United States Food and Drug Administration (FDA). Owing to their clinical benefits in metastatic settings, several CPIs are being investigated in perioperative settings. Therefore, we have discussed the current evidence of perioperative CPIs, and have summarized the results from recent trials in this section (Table 4).

### 4.1. Immunotherapy Alone

There have been two pivotal trials of neoadjuvant CPI alone to date [38,39]: The PURE-01 trial (NCT02736266) [38] was an open-label, single-arm, phase 2 study that assessed the activity of pembrolizumab as neoadjuvant immunotherapy before RC in patients with MIBC with predominant UC histology and cT2-3bN0 stage. Three cycles of pembrolizumab (200 mg every 3 weeks) were administered to patients before RC. The primary endpoint was pCR. A total of 92% of patients were eligible for cisplatin. Neoadjuvant pembrolizumab resulted in 42% pCR and 54% downstaging to non-muscle-invasive disease, and toxicity profiles were manageable. A total of 54.3% of patients with a combined positive score (CPS) ≥ 10 in programmed death-ligand 1 (PD-L1) expression showed pCR, but only 13.3% of patients with a PD-L1 CPS < 10 showed pCR. Recently, a study that evaluated the surgical safety of neoadjuvant pembrolizumab from the PURE-01 study population [40] indicated that high-grade complications (defined as Clavien –Dindo ≥ 3a) were observed in 34% of patients, and that there was no perioperative mortality at 90 days. Survival analysis from PURE-01 revealed that the pembrolizumab effect was maintained post-RC in most patients, with 1- and 2-year event-free survival (EFS) rates of 84.5% and 71.7%, respectively [41]. A statistically significant EFS benefit was observed in patients with a CR, and high PD-L1 CPS was significantly associated with longer EFS. The ABACUS trial (NCT02662309) [39,42]—a single-arm phase 2 study—investigated the efficacy and safety of two cycles of neoadjuvant atezolizumab (1200 mg every 3 weeks) prior to RC for MIBC (T2–4N0M0). The primary endpoint was pCR. Unlike PURE-01, all patients were ineligible for or refused cisplatin-based NAC. The rate of pCR and downstaging to non-muscle-invasive disease were 31% and 39%, respectively. Treatment-related grade 3/4 toxicity occurred in 12% of patients. Grade 3 or 4 surgical complications occurred in 31% of patients. Meanwhile, the combination of CPI therapy with other agents (other immunotherapy, chemotherapy, or other agents such as targeted therapy) in a neoadjuvant setting is being actively investigated.

### 4.2. Combination of Immunotherapy and Another Immunotherapy

Cytotoxic T-lymphocyte antigen 4 (CTLA-4), another key immune checkpoint, is expressed by activated T cells and regulatory T cells. Binding of CTLA-4 to its ligands (B7-1 (also known as CD80) and B7-2 (also known as CD86)) on antigen-presenting cells leads to inhibition of T cells [43]. Blocking of the T-cell negative regulator CTLA-4 allows CD28 and B7 interactions, which result in T-cell activation, proliferation, tumor infiltration and, ultimately, cancer cell death [44]. CTLA-4 inhibits the early activation and differentiation of T cells (typically in the lymph nodes), whereas programmed cell death protein 1 (PD-1) modulates their effector functions (mostly within tumors), which can lead to T-cell exhaustion [45]. Therefore, the combination of anti–PD-1 and CTLA-4 therapy triggers complementary mechanisms of therapeutic checkpoint inhibition [46]. In preclinical models, combined blockade of PD-1 and CTLA-4 achieved more pronounced antitumor activity than the blockade of either pathway alone [47,48]. Furthermore, the combination of CTLA-4 and PD-1/PD-L1 inhibitors showed promising clinical activity in several clinical trials [49]. In this context, a combination strategy of CTLA-4 and PD-1/PD-L1 inhibitors is being investigated extensively in neoadjuvant settings for MIBC.

The NABUCCO trial (NCT03387761)—a single-arm feasibility trial—assessed neoadjuvant ipilimumab/nivolumab combination therapy [50]. A total of 24 patients with stage III UC were treated with 3 mg/kg ipilimumab (day 1) plus 3 mg/kg ipilimumab, 1 mg/kg nivolumab (day 22), and 3 mg/kg nivolumab (day 43), followed by resection. The primary endpoint was feasibility to resect within 12 weeks of the start of treatment. A total of 96% of patients underwent resection within 12 weeks, and grade 3–4 immune-related adverse events (AEs) occurred in 55% of patients. Furthermore, a total of 46% of patients showed pCR, and 58% had no remaining invasive disease (pCR or pTisN0/pTaN0). DUTRENEO (NCT03472274) was a randomized phase 2 trial of durvalumab (DU)/tremelimumab (TRE) vs. chemotherapy in the neoadjuvant setting [51]. Cisplatin-eligible patients with cT2–T4aN0–1 were classified as immunologically “hot” or “cold” according to a tumor immune score devised by NanoString Technologies. Patients with “hot” tumors were randomized to three cycles of 1500 mg DU + 75 mg TRE every 4 weeks or standard cisplatin-based NAC. Patients in the “cold” arm received cisplatin-based NAC. The primary endpoint was pCR in the DU/TRE arm. In the “hot” arm, 36.4% of NAC and 34.8% of DU/TRE had a pCR, while 68.8% of patients in the NAC “cold” arm had a pCR. Grade 3–4 toxicities were more frequent in the NAC group. 

Until recently, combination treatment with PD-1/PD-L1 and CTLA-4 inhibitors has been studied extensively; however, newer strategies are now being investigated in the neoadjuvant setting, such as combination treatment with epacadostat, BMS-986205 (IDO-1 inhibitor), or NKTR-214/BEMPEG (CD122-preferential IL-2 pathway agonist). 

### 4.3. Combination of Immunotherapy and Cytotoxic Chemotherapy

Neoadjuvant immunotherapy with cytotoxic chemotherapy is being extensively investigated. Conventional chemotherapy can stimulate tumor-specific immune responses either by inducing immunogenic cell death (ICD) of tumor cells or by engaging immune effector mechanisms [52]. ICD, with several mechanisms—including exposure of calreticulin to the outer cell surface; release of adenosine triphosphate, annexin-1, and high-mobility group box 1 protein; autophagy; inflammasome activation; induction of type 1 interferon signaling, and release of mitochondrial formyl peptides—induces premortem reticular stress and releases tissue-damage-denoting substances (alarmins) that alert the immune system [53]. Furthermore, conventional chemotherapy can promote the activation of immune effector cells, hamper the functions of immunosuppressive cells, or alter whole-body physiology through the promotion and/or activation of mechanisms that ultimately support immunological competence [54]. These results provide a scientific rationale for the investigation of the combination of chemotherapy and immunotherapy; thus, many trials are currently ongoing.

BLASST-1 (NCT03294304) was a phase 2, single-arm trial that investigated the efficacy and safety of nivolumab with GC as neoadjuvant therapy for MIBC (cT2–T4aN ≤ 1 M0) [55]. Patients received four cycles of GC with nivolumab every 3 weeks, followed by RC within 8 weeks. The primary endpoint was PaR (≤ pT1N0); PaR was observed in 65.8% of patients, including those with N1 disease. The combination was safe, with manageable toxicities and no deaths from treatment. The majority of AEs were from GC; the overall rate of grade 3–4 AEs was 20%. One patient developed Guillain–Barré syndrome after surgery, which was resolved with intravenous immunoglobulin. There was no time delay to RC and no unexpected surgical complications from treatment. HCRN GU14-188 (NCT02365766) was a phase 1b/2 trial that evaluated the tolerability and efficacy of neoadjuvant GC with pembrolizumab in MIBC (cT2–4aN0M0) [56,57]. This study comprised two cohorts: cohort 1 was cisplatin-eligible, and cohort 2 was cisplatin-ineligible. Recently, the results of cohort 1 were reported: The primary endpoint was PaR (≤ pT1N0). PaR and pCR were observed in 61.1% and 44.4% of patients, respectively; PaR occurred in 53% of cT2 and 74% of cT3/4. The median time to RC from the last dose was 5.3 weeks. There was one death on post-RC day 9 due to mesenteric ischemia. At a median follow-up of 34.2 months, the estimated rates of 36-month relapse-free survival and OS were 63% and 82%, respectively. Currently, there are three ongoing phase 3 trials of neoadjuvant immunotherapy combined with cisplatin-based chemotherapy (NIAGARA [NCT03732677], ENERGIZE [NCT03661320], and KEYNOTE-866 [NCT03924856]), but their results have yet to be reported. Meanwhile, several ongoing studies are investigating immunotherapy with non-cisplatin-based chemotherapy, including nab-paclitaxel and gemcitabine as neoadjuvant treatments. 

### 4.4. Combination of Immunotherapy and Antibody–Drug Conjugates (ADCs)

ADCs are complex engineered therapeutics consisting of monoclonal antibodies directed toward tumor-associated antigens, to which highly potent cytotoxic agents are attached using chemical linkers [58]. Recently, several studies of ADCs in mUC have shown promising results.

Enfortumab vedotin is an ADC that comprises a fully human monoclonal antibody conjugated to a clinically validated microtubule-disrupting agent—monomethyl auristatin E—via a protease-cleavable linker [59]. Nectin-4—a type I transmembrane protein and member of a family of related immunoglobulin-like adhesion molecules—is known to be overexpressed in several epithelial cancers, especially bladder and breast cancer. However, the expression of nectin-4 in normal tissue is more limited. Enfortumab vedotin was able to bind to cell-surface-expressed nectin-4 with high affinity and induce cell death [60]. EV-201 (NCT03219333) is a global, phase 2, single-arm study that administered 1.25 mg/kg enfortumab vedotin (intravenously on days 1, 8, and 15 of every 28-day cycle) to patients with locally advanced or metastatic UC who were previously treated with platinum chemotherapy and anti–PD-1/PD-L1 therapy [59]. The primary endpoint was objective response rate (ORR). The confirmed ORR was 44%, including 12% with radiographic complete response (CR). Based on the results of EV-201, EV-301 (NCT03474107) was conducted; EV-301 is a global, open-label, phase 3 study that investigated enfortumab vedotin vs. chemotherapy in patients with locally advanced or metastatic UC who had previously received platinum-containing chemotherapy, and had disease progression during or after PD-1/PD-L1 inhibitor treatment [61]. Patients were randomly assigned to groups (1:1) to receive enfortumab vedotin or the investigator’s choice of standard docetaxel, paclitaxel, or vinflunine chemotherapy. The primary endpoint was OS. After an 11.1-month follow-up period, median OS was significantly prolonged by 3.9 months with enfortumab vedotin compared with chemotherapy (median OS: 12.9 vs. 9.0 months); similar OS benefits were observed in prespecified subgroups. PFS was also improved with enfortumab vedotin vs. chemotherapy (5.6 vs. 3.7 months). Both ORR and disease control rate were significantly higher with enfortumab vedotin vs. chemotherapy (40.6% vs. 17.9% and 71.9% vs. 53.4%, respectively; one-sided *p* < 0.001). Rates of treatment-related adverse events (TRAEs; 93.9% vs. 91.8%), including serious TRAEs (22.6% vs. 23.4%), were comparable between the enfortumab vedotin and chemotherapy groups. The FDA granted accelerated approval to enfortumab vedotin to treat patients with locally advanced or metastatic UC who previously received a PD-1/PD-L1 inhibitor and platinum-containing chemotherapy in the neoadjuvant/adjuvant, locally advanced, or metastatic settings. Based on these promising results, two randomized phase 3 trials of perioperative enfortumab vedotin with pembrolizumab vs. chemotherapy in cisplatin-eligible patients (NCT04700124)—or observation in cisplatin-ineligible patients (NCT03924895)—are ongoing in MIBC.

Sacituzumab govitecan is an ADC that recognizes trophoblast cell-surface antigen 2 (Trop-2)—a cell-surface glycoprotein highly expressed in aggressive bladder cancers. The antibody to Trop-2 is conjugated with a linker to a payload consisting of SN-38—the active metabolite of irinotecan [62,63]. Scott et al. performed a phase 1/2 basket study (NCT01631552) on patients with advanced solid tumors receiving intravenous sacituzumab govitecan administered on days 1 and 8 of 21-day cycles until progression or unacceptable toxicity, and reported the results of patients with mUC [64]. The ORR was 31% and the median PFS and OS were 7.3 and 18.9 months, respectively. TROPHY-U-01 (NCT03547973) is a multicohort, global, open-label phase 2 study evaluating the clinical activity of sacituzumab govitecan in patients with unresectable, locally advanced or metastatic UC. The results of cohort 1—which includes patients progressing after platinum and CPI therapy with unlimited prior lines of therapy—were recently reported [65]. The ORR, primary endpoint of this study, was 27%, including 5% with CR. Cohort 3 of this trial, which is actively accruing patients, is investigating the clinical benefit of sacituzumab govitecan in combination with pembrolizumab in the second-line setting [66]. On 13 April, 2021, the FDA granted accelerated approval to sacituzumab govitecan for patients with locally advanced or metastatic UC who previously received platinum-containing chemotherapy and either a PD-1 or a PD-L1 inhibitor. Based on the promising results of sacituzumab govitecan in the metastatic setting, the SURE trial has been planned; this trial is an open-label, sequential-arm, phase 2 study of neoadjuvant sacituzumab govitecan and sacituzumab govitecan plus pembrolizumab before RC for patients with MIBC who cannot receive or refuse cisplatin-based chemotherapy [67].

### 4.5. Combination of Immunotherapy and Other Emerging Agents or Radiotherapy

NEODURVARIB (NCT03534492) was a single-arm, phase 2 trial that assessed the impact of neoadjuvant durvalumab plus olaparib (a poly ADP-ribose polymerase inhibitor) in MIBC (cT2–T4aN0) [68]. Patients received 1500 mg durvalumab every 4 weeks for up to a maximum of 2 months (up to 2 doses/cycle) plus 300 mg olaparib for up to 56 days (two cycles of 28 days each cycle). The primary endpoint was pCR rate, which was 44.5%. One death related to postoperative complications was reported. Grade 3–4 AEs were detected in only 8.3% of patients. ABATE (NCT04289779) is an open-label, single-arm study to assess the efficacy and safety of cabozantinib (tyrosine kinase inhibitor whose targets include MET, AXL, and VEGFR2) with atezolizumab as neoadjuvant therapy for cT2–T4aN0/xM0 advanced UC patients who are either cisplatin-ineligible or decline cisplatin [69]. The primary endpoint is downstaging to non-muscle-invasive disease (< pT2). This study is ongoing, and results are not yet reported.

Genomic alterations in the oncogenic fibroblast growth factor receptor (FGFR) 3 pathway are well described in UC, and have led to extensive investigations of FGFR3-targeted therapies [70]. In the metastatic setting, several studies assessed the clinical benefit of FGFR inhibitors. The BLC2001 trial (NCT02365597)—an open-label phase 2 study—assessed the response in patients with locally advanced and unresectable or metastatic UC with FGFR alterations [71]. The confirmed response rate to erdafitinib therapy was 40% (3% with a CR and 37% with a partial response). Among the 22 patients who had undergone previous immunotherapy, the confirmed response rate was 59%. Currently, there are no studies of neoadjuvant FGFR-targeted agents with immunotherapy combination in muscle-invasive disease. There are some concerns about FGFR-targeted therapy being a proper partner of immunotherapy, as FGFR pathway activation is associated with non-T-cell-inflamed tumors in MIBC [72]. In the perioperative setting, only infigratinib (FGFR1–3-selective tyrosine kinase inhibitor) monotherapy is currently being investigated as neoadjuvant (NCT0422804) and adjuvant (NCT04197986) treatment for locally advanced UC.

Many efforts to find an appropriate partner for CPI therapy in the neoadjuvant setting are underway. There are several trials involving emerging agents—including CD73 inhibitor (NCT03773666), replication-competent oncolytic adenovirus (NCT04610671), and synthetic benzamide-derivative histone deacetylase inhibitor (NCT03978624). These studies are currently ongoing, and the results have not yet been reported.

Radiation can synergize with immunotherapy to improve oncological outcomes by causing ICD and increasing immune marker expression [73]. Based on this hypothesis, several trials of neoadjuvant immunotherapy with radiotherapy (RT) prior to cystectomy in MIBC are being conducted. RADIANT (NCT04543110) assesses the effect of sequential radiation and durvalumab immunotherapy given as treatment prior to surgery with RC for patients with bladder cancer who are unfit for or decline cisplatin. The RACE IT (NCT03529890, nivolumab + radiotherapy) and CIRTiN-BC (NCT04779489, several CPIs + radiotherapy) trials are also ongoing.

## 5. Adjuvant Immunotherapy in MIBC

There are three large-scale, randomized phase 3 trials for adjuvant immunotherapy (Table 5). The IMvigor 010 study (NCT02450331)—a multicenter, open-label, randomized phase 3 trial—evaluates atezolizumab for adjuvant therapy in patients with high-risk muscle-invasive UC (MIUC) [74]. Patients had ypT2–4a or ypN+ tumors following NAC or pT3–4a or pN+ tumors if no NAC was administered. Patients not treated with NAC must have been ineligible for or declined cisplatin-based NAC. A total of 807 patients were randomly assigned (1:1) to receive 1200 mg atezolizumab administered intravenously every 3 weeks for 16 cycles, up to one year, or to observation (whichever occurred first). A total of 6.6% patients had upper-tract MIUC. The primary endpoint was DFS; the trial did not meet its primary endpoint, with a nonsignificant difference in DFS between the atezolizumab group (19.4 months) and observation (16.6 months). Atezolizumab was generally tolerable, and had no new safety signals.

The CheckMate 274 trial (NCT02632409) is a recent randomized, double-blind, multicenter phase 3 trial of nivolumab vs. placebo in patients with high-risk MIUC after radical surgery, which reported positive results [75]. Patients had ypT2–4a or ypN+ tumors following NAC or pT3–4a or pN+ tumors if no NAC was administered. Patients were randomly (1:1) assigned to groups that received 240 mg nivolumab every 2 weeks or placebo for ≤1 year of adjuvant treatment. Patients had radical surgery within 120 days ± neoadjuvant cisplatin, or were ineligible for/declined cisplatin-based chemotherapy. The primary endpoints were DFS in all randomized patients and patients with tumor PD-L1 expression ≥1%. A total of 21% of patients had upper-tract UC. The primary endpoint of DFS was met in all randomized patients (median = 21.0 months for nivolumab, 10.9 months for placebo; hazard ratio (HR) = 0.70; *p* < 0.001) and in patients with PD-L1 ≥ 1% (median not reached for nivolumab, 10.8 months for placebo; HR = 0.53; *p* < 0.001). DFS improvement with nivolumab was generally consistent across subgroups. Grade 3–4 TRAEs occurred in 17.9% and 7.2% of patients in the nivolumab and placebo arms, respectively.

Although the IMvigor 010 (NCT02450331) and CheckMate 274 trials (NCT02632409) were similar in design, they showed conflicting results. These two trials had some differences in population and study design. CheckMate 274 included more upper tract disease than IMvigor 010 (6.6% in IMvigor 010 vs. 21% in CheckMate 274). In addition, CheckMate 274 was a placebo-controlled study, while IMvigor 010 was an observation-controlled study. In fact, the DFS of the experimental arm was similar in the two studies (19.4 months with atezolizumab (IMvigor 010) vs. 21.0 months with nivolumab (CheckMate 274)), but the DFS of their control groups showed a difference of approximately 6 months (16.6 months with observation (IMvigor 010) vs. 10.9 months with placebo (CheckMate 274)). Given that it is not appropriate to compare the two trials directly, these conflicting results should be interpreted cautiously. The AMBASSADOR (NCT03244384) trial—a multicenter, randomized phase 3 trial of adjuvant pembrolizumab vs. observation—is currently ongoing in patients with high-risk MIUC [76]. The results of this study are not yet reported.

Currently, the use of adjuvant immunotherapy with other agents is not being actively investigated. Instead of the “adjuvant-only” setting, adjuvant immunotherapy with other agents is being researched in conjunction with the neoadjuvant approach. We have summarized the major phase 3 trials involving perioperative (sequential) immunotherapy with other agents in Table 6.

## 6. Discussion

Perioperative CPI therapy, with or without other agents, has shown promising clinical efficacy and safety in several trials. However, several unanswered questions remain: The first is whether a perioperative CPI-based approach can replace perioperative cisplatin-based chemotherapy in cisplatin-eligible patients. The PURE-01 [38] and ABACUS [39] studies on single-agent neoadjuvant immunotherapy revealed pCR rates of 30–40%. Meanwhile, GETUG-AFU (NCT01812369) showed 35–45% pCR rates with cisplatin-based NAC [21] (Table 7). Considering these data, it is unclear whether single-agent neoadjuvant CPI therapy is superior to cisplatin-based NAC in patients who are fit for cisplatin. However, whether pCR is an optimal endpoint for neoadjuvant CPI trials is also not well established. The effects of immunotherapy can last longer than those of cytotoxic agents; therefore, long-term follow-up and survival results can provide a clinical impression about the efficacy of neoadjuvant immunotherapy in the future. Meanwhile, although adjuvant nivolumab showed clinical benefits in patients unfit for cisplatin in the adjuvant setting (patients who received cisplatin-based NAC were unfit for cisplatin in the adjuvant setting), it is unclear whether adjuvant nivolumab is superior to cisplatin in patients who are fit for cisplatin. Although it is also not appropriate to directly compare two different studies in terms of numerical HR difference, EORTC 30994 [22] exhibited a numerically lower HR (DFS, HR, 0.54 with GC vs. 0.70 with nivolumab) than CheckMate 274 [75]. However, CheckMate 274 exhibited a numerically comparable HR (0.53) in PD-L1-positive patients. 

However, the recent trend for perioperative trials is a combination strategy. Therefore, comparing the superiority of perioperative immunotherapy and chemotherapy may be inappropriate in this era of immunotherapy. It is necessary to check the results of the ongoing trials to determine whether a perioperative combination strategy can eventually become the standard of care.

The second issue associated with the perioperative setting is determining whether neoadjuvant or adjuvant treatment is appropriate. Although several guidelines prefer NAC over AC, there is no consensus on whether neoadjuvant or adjuvant immunotherapy is appropriate. Both approaches present potential advantages: Neoadjuvant treatment [77] enables less extensive surgery, response monitoring, and provides prognostic information, including pCR and surrogate markers. Furthermore, the neoadjuvant approach might be better tolerated than the adjuvant approach, owing to the relevant post-surgery morbidity that might prevent reasonable adjuvant treatment. Moreover, immunotherapy may be more appropriate in the neoadjuvant setting than in the adjuvant setting; this is because of the hypothesis that tumor-infiltrating lymphocytes mostly express the targets for immunotherapy; moreover, a higher load of tumor antigens is likely to be present when the primary tumor is still present for cross-priming at the time of immunotherapy [78]. However, the major concern for neoadjuvant treatment is delaying potentially curative RC in a proportion of patients—particularly those with symptoms and at risk of rapid clinical deterioration. According to the Keynote 052 trial [33], CPI monotherapy exhibited only 20% ORR in patients with metastatic disease who were treatment-naïve, and approximately 40% of patients were initially refractory to CPI therapy. Therefore, refractoriness to neoadjuvant immunotherapy is a major challenge that may interfere with treatment. Meanwhile, adjuvant treatment also has several advantages, including not delaying curative surgery, and treatment decision-making based on pathologic staging; however, adjuvant treatment cannot be delivered to patients who have had extensive surgeries, as it can increase the risk of postoperative complications and worsen performance status. Considering the pros and cons of these treatments, many ongoing perioperative trials are not confined to only the “one-side” strategy (neoadjuvant or adjuvant), and are being conducted with consecutive treatment strategies, ranging from neoadjuvant to adjuvant therapy.

Finally, a prognostic or predictive biomarker analysis may improve the efficacy of the perioperative strategy. There have been several efforts to investigate the molecular prognostic factors in MIBC. For example, p53 has been investigated as a prognostic marker of MIBC in several retrospective studies [79,80]. However, the evidence is lacking for the clinical utility of p53 as a prognostic marker. A previous trial, which investigated the prognostic impact of p53 in the adjuvant setting, failed to show the prognostic value of p53 [25]. The Cancer Genome Atlas (TCGA) project performed an integrated analysis of 131 UCs with whole genome/RNA sequencing, microarray, and reverse-phase protein array [81]; this study identified four clusters of UCs: Cluster I (“papillary like”) was enriched in tumors with papillary morphology and FGFR3 mutations/copy numbers/expression. Clusters I and II expressed high human epidermal growth factor receptor 2 (HER2; ERBB2) levels and an elevated estrogen receptor beta signaling signature. Cluster III (“basal/squamous-like”) was similar to that of basal-like breast cancers, as well as squamous-cell cancers of the head, neck, and lung. Until now, there are no definitive prognostic biomarkers in MIBC. However, given these results, future studies based on precision medicine should be performed. Currently, there are several ongoing molecular-biomarker-driven studies in the perioperative setting. 

Currently, there is no consensus on predictive biomarkers in the perioperative setting. Previous studies [28,30,31,33,35,82,83,84,85,86,87] have suggested several potential biomarkers of immunotherapy in the metastatic setting, including PD-L1 expression, molecular subtyping, tumor mutation burden, gene expression subtype, and DNA damage response gene alteration; however, the reliability of predictive biomarkers remains dubious in the perioperative and metastatic settings.

In the perioperative setting, the PURE-01 trial reported a significant nonlinear association between tumor mutation burden and pT0, and a cutoff at 15 mutations/Mb [38]. The ABACUS trial proposed that preexisting activated T cells were more prominent than expected, and correlated with better outcomes [42]. Additionally, IMvigor 011 suggested that post-surgical ctDNA positivity, which is associated with a high risk of recurrence and death, identified patients with MIUC that were likely to benefit from adjuvant atezolizumab [88]. Many ongoing trials are currently exploring potential biomarkers; we hope that these analyses will contribute to clinical improvement in patients who receive perioperative treatment.

## 7. Conclusions

Given the recurrence rates and poor outcomes of treatment with RC alone for MIBC, perioperative systemic treatment is important for improving the prognosis of patients with MIBC. Cisplatin-based perioperative chemotherapy is currently the primary perioperative strategy, and it is expected to be of significance in the future. Owing to the success of CPI therapy in advanced UC, perioperative immunotherapy has been extensively studied in MIBC. Perioperative immunotherapy has shown promising efficacy with relatively low toxicity. Furthermore, immunotherapy-based combination strategies have shown encouraging results. As novel agents including ADCs have recently shown promising results in UC, they are likely to be emerging options in the perioperative setting. We expect that the ongoing perioperative trials will achieve positive results and improve the prognosis of patients with MIBC.

## Figures and Tables

**Table 1 ijms-22-07201-t001:** Summary of trials for cisplatin-based neoadjuvant chemotherapy in muscle-invasive bladder cancer.

	SWOG-8710 [6]	BA06 30894 [8]	Choueiri et al. [13] (NCT00808639)	Plimack et al. [14] (NCT01031420)	Dash et al. [18]	MSK [20]
N	317	976	39	40	42	154
Phase	3	3	2	2	R	R
Regimen	MVAC	CMV	ddMVAC	aaMVAC	GC	GC
Duration of NAC, weeks	14	NA	8	6	12	12
Median time to definitive treatment after randomization, weeks	16	NA	14	9.7	19	17
Planned surgery rates, %	82	NA	97	98	NA	NA
pCR (pT0N0) rates, %	38	NA	26	38	26	21
Downstaging (<pT2) to non-muscle invasive disease, %	44	NA	49	53	36	46

R: retrospective; NA: not available.

**Table 2 ijms-22-07201-t002:** Summary of trials for cisplatin-based adjuvant chemotherapy in muscle-invasive bladder cancer.

	EORTC 30994, 2015 [22] (NCT00028756)	SOGUG, 2010 [24]	Cognetti et al., 2012 [23]
N	284	142	194
Phase	3	3	3
Regimen	GC, high-dose MVAC, MVAC	PGC	GC
DFS	5-year DFS rates: 47.6% (AC) vs. 31.8% (control)	NA	5-year DFS rates: 37.2% (AC) vs. 42.3% (control)
OS	5-year OS rates: 53.6% (AC) vs. 47.7% (control)	5-year OS rates: 60% (AC) vs. 31% (control)	5-year OS rates: 43.4% (AC) vs. 53.7% (control)

PGC: Paclitaxel/gemcitabine/cisplatin; NA, not available.

**Table 3 ijms-22-07201-t003:** Summary of trials for immune checkpoint inhibitors in advanced/metastatic urothelial carcinoma.

Trial	Drug	Treatment Line	Number	Phase	Primary Endpoint	ORR, %	Median OS, Months	Median PFS, Months	Grade 3–4 TRAE, %
IMvigor 210 [35] (NCT02108652)	Atezolizumab	1	119	2	ORR	23	15.9	2.7	16
IMvigor 211 [28] (NCT02302807)	Atezolizumab	3	467	3	OS	13	8.6	2.1	20
JAVELIN Solid Tumor [29] (NCT01772004)	Avelumab	2	249	1b	DLT	17	6.5	1.6	8
Study 1108 [34] (NCT01693562)	Durvalumab	2	191	1/2	Safety, ORR	18	18.2	1.5	7
CheckMate 275 [30,37] (NCT02387996)	Nivolumab	2	265	2	ORR	20	8.6	1.9	25
KETNOTE-052 [33,36] (NCT0233542)	Pembrolizumab	1	370	2	ORR	29	11.3	2.2	21
KETNOTE-045 [31,32] (NCT02256436)	Pembrolizumab	2	542	3	OS, PFS	21	10.1	2.1	15

DLT: Dose limiting toxicity.

**Table 4 ijms-22-07201-t004:** Summary of current neoadjuvant trials for immunotherapy with or without other agents in MIBC.

Trial	Agent	Phase	Population	Cisplatin Eligibility	Upper-Tract Disease Included
**Signle-Agent therapy**					
NCT02662309 (ABACUS)	Atezolizumab	2	cT2-T4N0	N	N
NCT02451423	Atezolizumab	2	cTa-T4N0	N	N
NCT03577132	Atezolizumab	2	cT2-T4N0-1	Y	N
NCT03498196 (BL-AIR)	Avelumab	1/2	cT2-T4aN0	N	N
NCT03406650 (SAKK 06/17)	Durvalumab	2	cT2-T4N0-1	Y	Y
NCT02736266 (PURE-01)	Pembrolizumab	2	cT2-T4N0	Y	N
NCT03212651 (PANDORE)	Pembrolizumab	2	cT2-T4N0	N	N
NCT03319745	Pembrolizumab	2	cT2-T4N0	Y	N
**CPI with other immunotherapy**					
NCT02812420	Durvalumab + Tremelimumab	1	cT2-3aN0	Y	Y
NCT03472274 (DUTRENEO)	Durvalumab + Tremelimumab	2	cT2-T4N0-1	Y	N
NCT03234153 (NITIMIB)	Durvalumab + Tremelimumab	2	cTa-T4anyN	N	N
NCT02845323	Nivolumab + Urelumab	2	cTa-T4N0	N	N
NCT03387761 (NABUCCO)	Nivolumab + Ipilimumab	1b	cTa-T4anyN	Y	N
NCT03520491 (CA209-9DJ)	Nivolumab + Ipilimumab	2	cT2-4aN0	N	N
NCT03532451 (PrE0807)	Nivolumab + Lirilumab	1b	cT2-T4aN0-1	Y	N
NCT04209114 (CA045-009)	Nivolumab + Bempeg	3	cT2-T4N0	N	N
NCT03832673 (PECULIAR)	Pembrolizumab + Epacadostat	2	cT2-T3N0	Y	N
NCT04586244 (Optimus)	Retifanlimab + Epacadostat	2	cT2-T3bN0	N	N
NCT04430036	Zalifrelimab + Balstilimab	2	cT2-T4N0-1	Y	N
**CPI with chemotherapy**					
NCT02989584	Atezolizumab + GC	2	cT2-T4aN0	Y	N
NCT03674424 (AURA)	Avelumab + Chemotherapy	2	cT2-T4anyN	Y	N
NCT03732677 (NIAGARA)	Durvalumab + GC	3	cT2-T4aN0	Y	N
NCT03549715 (NEMIO)	Durvalumab + Tremelimumab + ddMVAC	1/2	cT2-T4N0-1	Y	N
NCT03912818	Durvalumab + Chemotherapy	2	cT2-T4N0-1	Y	N
NCT03661320 (ENERGIZE)	Nivolumab + BMS-986205 + GC	3	cT2-T4N0	Y	N
NCT03294304 (BLASST-1)	Nivolumab + GC	2	cT2-T4N0-1	Y	N
NCT03558087	Nivolumab + GC	2	cTa-T4N0	Y	N
NCT04506554	Nivolumab + aaMVAC	2	cT2-T3N0	Y	N
NCT04383743	Pembrolizumab + MVAC	2	cT2-T4N0-1	Y	N
NCT02690558	Pembrolizumab + GC	2	cT2-T4N0	Y	N
NCT02365766 (HCRN GU14-188)	Pembrolizumab + GC	2	cT2-T4N0	Y/N (two cohorts)	Y
NCT03924856 (KEYNOTE-866)	Pembrolizumab + GC	3	cT2-T4N0-1	Y	N
NCT04861584 (GZZJU-2021NB)	Teriprizumab + GC	2	cT2-T4N0-1	Y	N
NCT04730219	Tislelizumab + Nab-paclitaxel	2	cT2-T4aN0	Y	N
NCT04553939	Toripalimab + Gemcitabine	2	cT2-T4anyN	N	N
NCT04099589	Toripalimab + GC	2	cT2-T4aN0	Y	Y
**CPI with other agents**					
NCT04289779 (ABATE)	Atezolizumab + Cabozantinib	2	cT2-T4anyN	N	N
NCT03534492 (NEODURVARIB)	Durvalumab + Olaparib	2	cT2-T4aN0	Y	N
NCT03773666 (BLASST-2)	Durvalumab + Oleclumab	1	cT2-T4aN0	N	N
NCT04610671	Nivolumab + CG0070	1	cT2-T4aN0	N	N
NCT03518320	Nivolumab + TAR-200	1	cT2-T3N0-1	N	N
NCT04700124 (KEYNOTE-B15/EV-304)	Pembrolizumab + Enfortumab vedotin	3	cT2-T4N0-1	Y	N
NCT03924895 (KEYNOTE-905/EV-303)	Pembrolizumab + Enfortumab vedotin	3	cT2-T4N0-1	N	N
NCT03978624	Pembrolizumab + Entinostat	2	cT2-T4aN0	N	N
NA (SURE)	Pembrolizumab + Sacituzumab govitecan	2	cT2-T4N0	N	N
NCT04813107	Tislelizumab + APL-1202	1/2	cT2-T4aN0	N	N
**CPI with radiation**					
NCT04543110 (RADIANT)	Durvalumab + Radiation	2	cT2-T4aN0	N	N
NCT04779489 (CIRTiN-BC)	CPIs + Radiation	2	anyTN+	N	N
NCT03529890 (RACE IT)	Nivolumab + Radiation	2	cT3-T4anyN	N	N

**Table 5 ijms-22-07201-t005:** Summary of phase 3 trials for adjuvant immunotherapy in muscle-invasive bladder cancer.

Trial	Phase	Agent	Control	N	Primary Endpoint	Upper Tract	Cisplatin-Based NAC
NCT03244384 [76] (AMBASSADOR)	3	Pembrolizumab	Observation	739	OS, DFS	Included	Included
NCT02632409 [75] (CheckMate 274)	3	Nivolumab	Placebo	700	DFS	Included	Included
NCT02450331 [74] (IMvigor010)	3	Atezolizumab	Observation	809	DFS	Included	Included

**Table 6 ijms-22-07201-t006:** Summary of phase 3 trials for perioperative sequential treatment.

Trial	Phase	Agent	Arm
NCT04209114 (CA045-009)	3	Nivolumab + Bempeg	Arm A: Neoadjuvant nivolumab + bempeg => RC => Adjuvant nivolumab + bempeg Arm B: Neoadjuvant nivolumab => RC => Adjuvant nivolumab Arm C: RC alone
NCT03732677 (NIAGARA)	3	Durvalumab + GC	Arm A: Neoadjuvant durvalumab + GC => RC => Adjuvant durvaluamb Arm B: Neoadjuvant GC => RC => No adjuvant therapy
NCT03661320 (ENERGIZE)	3	Nivolumab + BMS-986205 + GC	Arm A: Neoadjuvant GC => RC => No adjuvant therapy Arm B: Neoadjuvant nivolumab + placebo + GC => RC => Adjuvant nivolumab + placebo Arm C: Neoadjuvant nivolumab + BMS-986205 + GC => RC => Adjuvant nivolumab + BMS-986205
NCT03924856 (KEYNOTE-866)	3	Pembrolizumab + GC	Arm A: Neoadjuvant pembrolizumab + GC => RC => Adjuvant pembrolizumab Arm B: Neoadjuvant placebo + GC => RC => Adjuvant placebo
NCT04700124 (KEYNOTE-B15/EV-304)	3	Pembrolizumab + Enfortumab vedotin	Arm A: Neoadjuvant pembrolizumab + enfortumab vedotin => RC => Adjuvant pembrolizumab + enfortumab vedotin Arm B: Neoadjuvant GC => RC => No adjuvant therapy
NCT03924895 (KEYNOTE-905/EV-303)	3	Pembrolizumab + Enfortumab vedotin	Arm A: Neoadjuvant pembrolizumab + enfortumab vedotin => RC => Adjuvant pembrolizumab + enfortumab vedotin Arm B: Neoadjuvant pembrolizumab => RC => Adjuvant pembrolizumab Arm C: RC alone

**Table 7 ijms-22-07201-t007:** Summary of pathologic response in major neoadjuvant trials.

	Pure-01 [38,41]	ABACUS [39,42]	NABUCCO [50]	DUTRENEO [51]	BLASST-1 [55]	HCRN GU14-188 [56,57]		SWOG8710 [6]	GETUG/AFU [21]	
	PEM	ATEZO	IPI/NIVO	DU/TREME	NIVO + GC	Cohort 1: PEM + GC	Cohort 2: PEM + GEM	MVAC	Arm A: ddMVAC	ARM B: GC
N	143	88	24	23	41	43	37	317	248	245
pCR (%)	39	31	46	35	49	44	45	38	42	36
Downstaging (<pT2) (%)	56	39	58	57	66	61	52	NA	63	49

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
