# Peer review of "Perioperative Systemic Treatment for Muscle-Invasive Bladder Cancer: Current Evidence and Future Perspectives"

_ijms, 2021, doi:10.3390/ijms22137201_

Round 1

Reviewer 1 Report

This is a very comprehensive review of the very important subject. Although a lot of scientific literature dedicated to this subject, some aspects of the approach is well described here. I believe that this review can be published. 

Author Response

Reviewer #1

This is a very comprehensive review of the very important subject. Although a lot of scientific literature dedicated to this subject, some aspects of the approach is well described here. I believe that this review can be published.

  • Response from authors: Thank you for your comments. Perioperative systemic treatment is expected to be of significance in the treatment of muscle-invasive bladder cancer in the future. We hope that our article will provide this clinical impression to readers. Thank you for your careful review.

Reviewer 2 Report

The article entitled 'Perioperative Systemic Treatment for Muscle-Invasive Bladder Cancer: Current Evidence and Future Perspectives' is based on a systematic review of databases selected by the authors. It interestingly discusses current and future options and perspectives for neoadjuvant and adjuvant treatment in patients with MIBC.An important part of the article is the one concerning neoadjuvant therapy with FDA-approved drugs: atezolizumab, avelumab, durvalumab, nivolumab, and pembrolizumab, but also the use of immunotherapy with another immunotherapy, immunotherapy with cytotoxic chemotherapy, immunotherapy with antibody-drug conjugates (ADCs) and immunotherapy with other emerging agents or radiotherapy. 
In the next, less extensive part of the article, the authors address the issue of adjuvant immunotherapy in MIBC. The authors note that clinical trials on the use of immunotherapy after surgery are less intensively conducted.
In the section entitled: discussion, current and future problems related to the selection of the best course of action in patients with MIBC are presented.
In my opinion, this review paper is interesting and summarises the results of the most interesting and advanced clinical trials to date.
It is worth publishing because of the relevance of the subject matter, the comprehensiveness and the systematic way in which it has been addressed.

Author Response

Reviewer #2

The article entitled 'Perioperative Systemic Treatment for Muscle-Invasive Bladder Cancer: Current Evidence and Future Perspectives' is based on a systematic review of databases selected by the authors. It interestingly discusses current and future options and perspectives for neoadjuvant and adjuvant treatment in patients with MIBC. An important part of the article is the one concerning neoadjuvant therapy with FDA-approved drugs: atezolizumab, avelumab, durvalumab, nivolumab, and pembrolizumab, but also the use of immunotherapy with another immunotherapy, immunotherapy with cytotoxic chemotherapy, immunotherapy with antibody-drug conjugates (ADCs) and immunotherapy with other emerging agents or radiotherapy.

In the next, less extensive part of the article, the authors address the issue of adjuvant immunotherapy in MIBC. The authors note that clinical trials on the use of immunotherapy after surgery are less intensively conducted.

In the section entitled: discussion, current and future problems related to the selection of the best course of action in patients with MIBC are presented.

In my opinion, this review paper is interesting and summarises the results of the most interesting and advanced clinical trials to date.

It is worth publishing because of the relevance of the subject matter, the comprehensiveness and the systematic way in which it has been addressed.

  • Response from authors: Thank you for your comments. Perioperative systemic treatment is expected to be of significance in the treatment of muscle-invasive bladder cancer in the future. We hope that our article will provide this clinical impression to readers. Thank you for your careful review.

Reviewer 3 Report

This manuscript entitled “Perioperative Systemic Treatment for Muscle-Invasive Bladder Cancer: Current Evidence and Future Perspectives” provides an organized review to represent the perioperative systemic treatment for MIBC. However, this article seems suitable for clinical application due to lacking the view point of molecular sciences.

  1. Briefly, the text focused on the clinical efficacy of perioperative systemic treatment but not on the molecular interactions, e.g., the cross-talk between different immunotherapy or immunotherapy/chymotherapy.
  2. The authors should review some molecular markers or array for evaluating the prognostic result; it would make the article valuable.
  3. In Discussion, the English grammar should be re-edited to polish this article.

Author Response

Reviewer #3

This manuscript entitled “Perioperative Systemic Treatment for Muscle-Invasive Bladder Cancer: Current Evidence and Future Perspectives” provides an organized review to represent the perioperative systemic treatment for MIBC. However, this article seems suitable for clinical application due to lacking the view point of molecular sciences.

  1. Briefly, the text focused on the clinical efficacy of perioperative systemic treatment but not on the molecular interactions, e.g., the cross-talk between different immunotherapy or immunotherapy/chemotherapy.
  • Response from authors: We thank the reviewer for this comment, with which we agree. Therefore, we have revised the manuscript to include text on molecular interactions between checkpoint inhibitors and other immunotherapies (IOs) or chemotherapy. In fact, we have considered other molecular interactions such as CPI with other novel therapies (e.g., PARP inhibitors, antibody-drug conjugates, and radiotherapy). However, an excessively detailed revision on other molecular interactions does not fit the aim of our review. However, currently, IO with another IO and IO with chemotherapy combinations are being extensively studied in several clinical trials and these combinations are of great interest. Thus, we have revised our manuscript according to your comments. The revised contents are highlighted in yellow in line 158-172, and 195-208. Thank you.

  1. The authors should review some molecular markers or array for evaluating the prognostic result; it would make the article valuable.
  • Response from authors: We thank the reviewer's valuable comment and completely agree with the comment. Thus, we have added text highlighted in yellow regarding molecular prognostic factors in line 429-445. We think this added information makes our article valuable.

  1. In Discussion, the English grammar should be re-edited to polish this article.
  • Response from authors: Thank you for comment. We have re-edited the entire article. We have attached the “certificate of grammar editing by Editage”.

Reviewer 4 Report

The article is well written and well structured and summarizes current knowledge in the field. The discussion is well structured and focuses on the main issues that have been covered. The manuscript can be published

Author Response

Reviewer #4

The article is well written and well structured and summarizes current knowledge in the field. The discussion is well structured and focuses on the main issues that have been covered. The manuscript can be published

Response from authors: Thank you for your comments. Perioperative systemic treatment is expected to be of significance in the treatment of muscle-invasive bladder cancer in the future. We hope that our article will provide this clinical impression to readers. Thank you for your careful review. 

Round 2

Reviewer 3 Report

This manuscript is required to check spell carefully before publication.